# RAG-3DSG: Enhancing 3D Scene Graphs with Re-Shot Guided Retrieval-Augmented Generation

## Abstract

Open-vocabulary 3D Scene Graph (3DSG) generation can enhance various downstream tasks in robotics, such as manipulation and navigation, by leveraging structured semantic representations. A 3DSG is constructed from multiple images of a scene, where objects are represented as nodes and relationships as edges. However, existing works for open-vocabulary 3DSG generation suffer from both low object-level recognition accuracy and speed, mainly due to constrained viewpoints, occlusions, and redundant surface density. To address these challenges, we propose **RAG-3DSG** to mitigate aggregation noise through re-shot guided uncertainty estimation and support object-level Retrieval-Augmented Generation (RAG) via reliable low-uncertainty objects. Furthermore, we propose a dynamic downsample-mapping strategy to accelerate cross-image object aggregation with adaptive granularity. Experiments on Replica dataset demonstrate that RAG-3DSG significantly improves node captioning accuracy in 3DSG generation while reducing the mapping time by two-thirds compared to the vanilla version.

## 1 INTRODUCTION

Compact and expressive representation of complex and semantic-rich 3D scenes has long been a fundamental challenge in robotics, with direct impact on downstream tasks such as robot manipulation (Shridhar et al., 2022; Rashid et al., 2023) and navigation (Gadre et al., 2022; Shah et al., 2023). A promising solution is 3D scene graphs (3DSGs) (Armeni et al., 2019; Gay et al., 2018), which encode a scene into a graph where nodes denote objects and edges capture their pairwise relationships. Early efforts focus on building 3DSGs by detecting objects and relationships from a closed vocabulary (Hughes et al., 2022; Rosinol et al., 2021; Wu et al., 2021). While these methods perform well and are efficient in fixed environments, their reliance on a closed vocabulary restricts their generalization to unseen scenes in the real-world. To mitigate this limitation, recent approaches (Gu et al., 2024; Werby et al., 2024; Koch et al., 2024; Maggio et al., 2024; Jatavallabhula et al., 2023) leverage foundation models to provide open-vocabulary 3DSG generation, which produces more expressive representations for more diverse scenes.

Despite this progress, open-vocabulary approaches largely adhere to the pre-defined one-way pipeline of per-image object-level information extraction followed by cross-image aggregation. However, as illustrated in the upper part of Figure 1, constrained viewpoints, occlusions, and other poor imaging conditions can introduce significant level of noise to object-level information extraction and reduces the accuracy of cross-image aggregation. For example, as shown in Figure 1(b), although the object of interest is the table, the presence of an occluding vase leads to multiple crop captions being mistakenly recognized as vase. When aggregating multiple captions or embeddings across images, such misleading semantics can compromise the accuracy of the resulting 3DSGs. Such noise-induced inaccuracies in 3DSGs are unacceptable for downstream robotic tasks, particularly in safety-critical scenarios. For example, if a medication bottle is incorrectly described as a beverage container, a service robot could deliver dangerous substances to humans. Therefore, it is crucial to assess the uncertainty in per-image object-level information and mitigate the noise accordingly. At the same time, we observe that existing methods use object crops only and miss the comprehensive descriptors of the target object and also the broader contextual information, which provide valuable clues for foundation models' captioning.

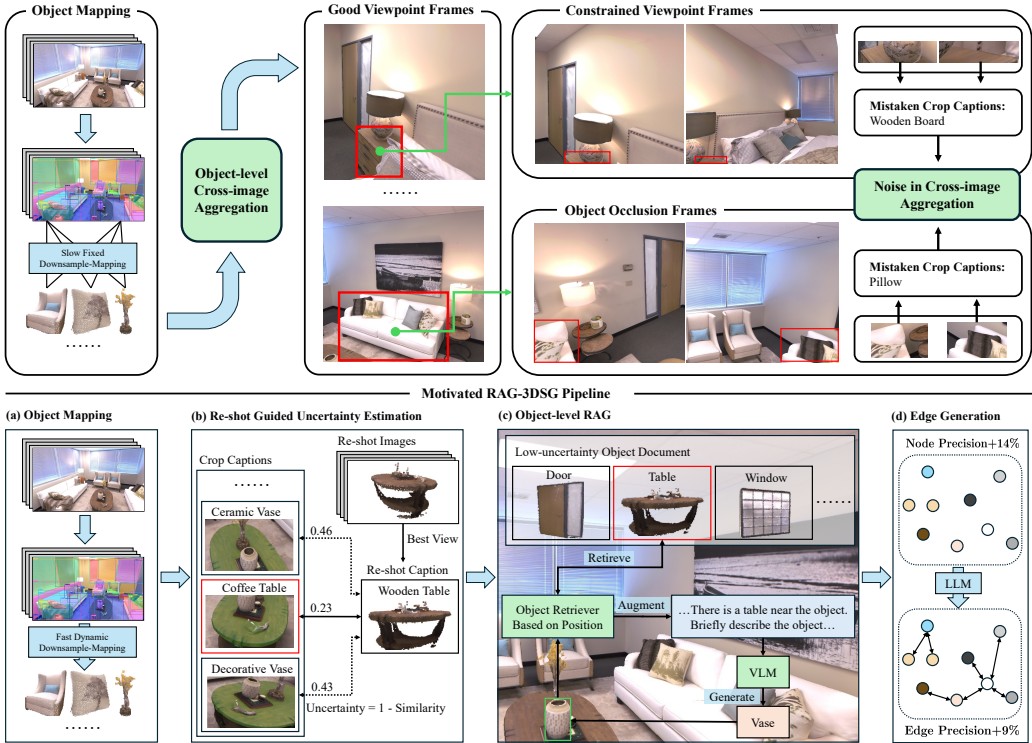

Figure 1: An overview of the RAG-3DSG framework. The upper part illustrates common challenges in multi-view 3D scene graph generation. Our pipeline addresses these issues through (a) Multi-view RGB-D frames are segmented and fused into a global object list with point clouds and semantic embeddings. (b) Re-shot images are used to select the best-view caption, which is compared with crop captions to estimate uncertainty; clustering is applied and the top-1 cluster is retained for fusion. (c) Low-uncertainty objects form a retrieval document, while high-uncertainty objects leverage retrieved context for caption refinement via a VLM. (d) Finally, spatial and semantic relationships among objects are inferred by an LLM to construct the 3D scene graph.

To address these challenges, we propose **RAG-3DSG**, a 3DSG generation framework that mitigates noise in aggregation through re-shot guided uncertainty estimation. It treats the 3DSG under construction as a database, and use the principle of Retrieval-Augmented Generation (RAG) to leverage surrounding context to enhance the representation of objects with high uncertainty. Specifically, we first adopt a dynamic downsample-mapping strategy to construct a global 3D object list with low computational cost (Figure 1a). Inspired by the concept of optimal viewpoint (Vázquez et al., 2001), our method then renders the object point clouds reconstructed from multiple images to obtain the best-view re-shot images. Next, we perform uncertainty estimation by comparing object captions generated from these re-shot images with those from the crop images (Figure 1b). Low-uncertainty objects are directly used to construct an object document, while high-uncertainty objects will trigger the retrieval of surrounding high certainty object documents on the 3DSG to guide caption refinement (Figure 1c), iteratively improve the accuracy of 3DSG.

Experiments on Replica (Straub et al., 2019) dataset demonstrate that our method consistently outperforms existing baselines, achieving an average node precision of 0.82 (vs. 0.68 for ConceptGraphs (Gu et al., 2024)) and edge precision of 0.91 (vs. 0.85 for ConceptGraphs-Detector (Gu et al., 2024)), while maintaining more valid objects and edges. In addition, our dynamic downsample-mapping strategy reduces the time of 3D object mapping by nearly two-thirds compared to the fixed downsample-mapping strategy in ConceptGraphs (Gu et al., 2024) (from 6.65s/iter to 2.49s/iter when the base voxel size is set to 0.01 on Replica Room 0), demonstrating improved accuracy, robustness, and efficiency in 3DSG construction.

## 2 RELATED WORK

**Scene Graph Generation (SGG)** Scene graphs were initially introduced in the 2D domain to extract objects and their relationships from images, providing a structured representation that supports a highly abstract understanding of scenes for intelligent agents (Sun et al., 2023; Liu et al., 2021; Yin et al., 2018; Krishna et al., 2017; Lu et al., 2016; Johnson et al., 2015). A scene graph is typically composed of three fundamental elements—objects, attributes, and relations—and can be expressed as a set of visual relationship triplets in the form of <subject, relation, object>(Li et al., 2024). In the field of 3D scene representations, several works (Wald et al., 2020; Kim et al., 2019; Armeni et al., 2019) have drawn inspiration from 2D scene graphs and extended the concept into the 3D domain. Compared with original 3D scene representations such as point clouds with per-point semantic vector, which are often overly dense and difficult to interpret, 3D scene graphs (3DSGs) provide a more compact and structured abstraction of the scene. By organizing objects and their relationships into a graph representation, they enable more efficient reasoning and facilitate downstream tasks such as robotic navigation (Gadre et al., 2022; Shah et al., 2023) and scene understanding (Rana et al., 2023; Agia et al., 2022). Early efforts in 3D scene graph generation (3DSGG) enabled the construction of real-time systems capable of dynamically building hierarchical 3D scene representations (Hughes et al., 2022; Rosinol et al., 2021; Wu et al., 2021). However, these methods were confined to the closed-vocabulary setting, which restricted their applicability to a limited range of tasks. More recently, several works (Gu et al., 2024; Werby et al., 2024; Koch et al., 2024; Maggio et al., 2024; Jatavallabhula et al., 2023) have begun to explore open-vocabulary approaches for 3D scene graph generation. Using Vision Language Models (VLMs) and Large Language Models (LLMs), these methods sacrifice part of the real-time capability but significantly expand the range of object categories and relations that can be recognized, thus broadening the applicability to a wider variety of downstream tasks.

**Open-vocabulary 3DSGG** Recent advances have extended 3DSGG to the open-vocabulary setting by leveraging VLMs and LLMs. The existing approaches can be divided into two main paradigms. The first paradigm follows a caption-first strategy, where objects in each image are independently described and later aggregated into scene-level semantics. In practice, multiple masked views of the same object are captioned separately, and a LLM is then used to aggregate these descriptions into a final caption (e.g., Gu et al. (2024)). The second paradigm adopts an embedding-first strategy, where semantic embeddings of multiple masked views of the same object are first extracted and aggregated (through like weighted fusion) without explicit captioning. The aggregated embeddings are then converted into captions or directly aligned with user queries in a joint vision–language space using models such as CLIP (Werby et al., 2024; Koch et al., 2024; Jatavallabhula et al., 2023). In addition, some works assign captions to object embeddings by matching them against an arbitrary vocabulary with CLIP, a strategy commonly referred to as open-set 3DSG (Maggio et al., 2024). Despite their differences, both paradigms share the same underlying pipeline of per-image information extraction followed by cross-image information aggregation. However, the information extraction process is often affected by factors such as constrained viewpoints and object occlusion, which introduce noise into the representations. As a result, aggregated information may be inaccurate, leading to reduced precision in the node and edge captions of the 3DSGs. This limitation cannot be simply resolved by adopting more powerful captioning models, as the noise originates from inherent challenges in multi-view perception such as occlusion and viewpoint constraints.

**Time Considerations in 3DSGG** Closed-vocabulary 3D scene graph generation methods are typically lightweight and efficient, since they operate on a fixed set of semantic categories. This efficiency enables some approaches to support online (Wu et al., 2023) or even faster-than-real-time construction (e.g., Hou et al. (2025) achieves 7 ms end-to-end latency). In contrast, open-vocabulary scene graph generation relies on VLMs or LLMs to provide flexible semantics, which introduces significant runtime overhead. As a result, existing open-vocabulary methods (Gu et al., 2024; Koch et al., 2024) are usually performed offline, making real-time deployment challenging. Unlike strictly open-vocabulary methods, Maggio et al. (2024) adopts an open-set formulation, which relaxes category constraints without relying on heavy VLM/LLM inference. This design makes real-time 3DSGG feasible, whereas open-vocabulary approaches typically remain offline. Even in the offline setting, construction time remains a critical bottleneck: in addition to the heavy time cost of vision-language reasoning, cross-image aggregation also incurs significant latency, further limiting the practicality of open-vocabulary 3DSGG.

## 3 METHOD

In this section, we present our proposed framework RAG-3DSG for open-vocabulary 3D scene graph generation. The overall pipeline is illustrated in Figure 1. Our pipeline consists of three main stages: (1) **Cross-image Object Mapping** (Section 3.1), where we perform 2D segmentation, dynamic downsampling, and 3D object fusion to construct a global object list with object-level point clouds and semantic embeddings; (2) **Node Caption Generation** (Section 3.2), where we first obtain initial captions, then perform re-shot guided uncertainty estimation, and finally refine high-uncertainty objects using object-level RAG; (3) **Edge Caption Generation** (Section 3.3), where we use a structured LLM prompt to produce interpretable relationship captions.

### 3.1 CROSS-IMAGE OBJECT MAPPING

Given an RGB-D image sequence $I = \{I_1, I_2, \ldots, I_t\}$, each image is represented as $I_i = \langle I_i^{\mathrm{RGB}}, I_i^{\mathrm{D}}, P_i \rangle$, where $I_i^{\mathrm{RGB}}$, $I_i^D$, and $P_i$ denote the color image, depth image, and camera pose respectively. Each image $I_i$ yields a set of detected objects $O_i^{\mathrm{local}} = \{o_{i,1}^{\mathrm{local}}, \ldots, o_{i,M}^{\mathrm{local}}\}$ obtained from segmentation. Our objective is to construct a global object list $O_t^{\mathrm{global}} = \{o_{t,1}^{\mathrm{global}}, \ldots, o_{t,N}^{\mathrm{global}}\}$ by incrementally fusing the per-image objects $O_t^{\mathrm{local}}$ into the accumulated set $O_{t-1}^{\mathrm{global}}$. Following Gu et al. (2024), we adopt incremental object-level cross-image mapping and further introduce a dynamic downsample-mapping strategy to improve efficiency and effectiveness.

#### 3.1.1 2D SEGMENTATION

For $t$-th input image $I_t^{\mathrm{RGB}}$, we first extract local object-level information. A class-agnostic segmentation model $\mathrm{SAM}(\cdot)$ (Kirillov et al., 2023) produces a set of 2D object masks $\{m_{t,i}\}_{i=1\ldots M} = \mathrm{SAM}(I_t^{\mathrm{RGB}})$. Next, a visual semantic encoder $\mathrm{CLIP}(\cdot)$ (Radford et al., 2021) is used to obtain semantic embeddings $\{f_{t,i}\}_{i=1\ldots M} = \mathrm{CLIP}(I_t^{\mathrm{RGB}}, \{m_{t,i}\}_{i=1\ldots M})$ for each masked region. To prepare for 3D information aggregation, the camera pose $P_t$ is used to project each 2D mask into 3D space, producing a set of point clouds $\{p_{t,i}\}_{i=1\ldots M}$. With these extracted information, we define the local object list of $I_t$ as $O_t^{\mathrm{local}} = \{o_{t,1}^{\mathrm{local}}, \ldots, o_{t,M}^{\mathrm{local}}\}$, where $o_{t,i}^{\mathrm{local}} = \langle f_{t,i}^{\mathrm{local}}, p_{t,i}^{\mathrm{local}} \rangle$. This local object list $O_t^{\mathrm{local}}$ serves as the basis for subsequent 3D object mapping and fusion.

#### 3.1.2 DYNAMIC DOWNSAMPLING

Before performing object mapping for $o_{t,i}^{\mathrm{local}}$ with point clouds $p_{t,i}$ and semantic embeddings $f_{t,i}$, we downsample the point clouds to reduce computational cost. Existing approaches typically adopt a fixed voxel size $\delta^{\mathrm{sample}}$, which is determined by the size of smaller objects within the scene. This strategy has a clear drawback that large objects remain lack-sampled, resulting in unnecessarily dense point clouds. To address this issue, we propose a dynamic downsampling strategy that adapts the voxel size according to the scale of each object. This not only improves efficiency but also facilitates the subsequent re-shot guided uncertainty estimation (Section 3.2.2) by ensuring that object pixels in re-shot images are dense enough to faithfully capture the underlying object. Formally, the voxel size $\delta_{t,i}^{\mathrm{voxel}}$ for the point cloud $p_{t,i}$ of object $o_{t,i}^{\mathrm{local}}$ is defined as:

$$\delta_{t,i}^{\mathrm{voxel}} = \delta^{\mathrm{sample}} \cdot \|\mathrm{Bbox}(p_{t,i})\|_2^{1/2}, \tag{1}$$

where $\delta^{\mathrm{sample}}$ denotes the fixed base voxel size, and $\|\mathrm{Bbox}(p_{t,i})\|_2$ corresponds to the Euclidean norm of the bounding box size vector, i.e., the diagonal length of the 3D bounding box, which reflects its overall spatial extent. For simplicity, we continue to use $p_{t,i}$ to denote the downsampled point cloud in the following sections. Compared with fixed voxel-size downsampling, our strategy yields denser point clouds for smaller objects and sparser ones for larger objects within the scene, thereby simultaneously achieving both finer granularity and higher efficiency, without incurring the usual trade-off between the two. Concrete examples of dynamic voxel sizes for common indoor objects are provided in Appendix A.2.

### 3.1.3 3D OBJECT FUSION

Incremental object mapping and fusion begins after dynamic downsampling. We use the local object list $O_1^{\text{local}}$ in the first image to initialize the global object list as $O_1^{\text{global}} = O_1^{\text{local}}$. Later, for each object $o_{t,i}^{\text{local}} = \langle f_{t,i}^{\text{local}}, p_{t,i}^{\text{local}} \rangle$ in the $t$-th image input, we follow Gu et al. (2024) to construct a fusion similarity $\theta(i,j)$ as follows,

$$\theta(i,j) = \theta_{\text{semantic}}(i,j) + \theta_{\text{spatial}}(i,j), \tag{2}$$

$\theta_{\text{semantic}}(i,j)$ is the semantic similarity between $o_{t,i}^{\text{local}}$ and $o_{t-1,j}^{\text{global}}$ as follows,

$$\theta_{\text{semantic}}(i,j) = (f_{t,i}^{\text{local}})^T f_{t-1,j}^{\text{global}}/2 + 1/2, \tag{3}$$

and $\theta_{\text{spatial}}(i,j)$ is the spatial similarity between $o_{t,i}^{\text{local}}$ and $o_{t-1,j}^{\text{global}}$ as follows,

$$\theta_{\text{spatial}}(i,j) = \text{dnnratio}(p_{t,i}^{\text{local}}, p_{t-1,j}^{\text{global}}), \tag{4}$$

where $\text{dnnratio}(\cdot)$ is the proposed dynamic nearest neighbor ratio, equal to the proportion of points in point cloud $p_{t,i}^{\text{local}}$ that have nearest neighbors in point cloud $p_{t-1,j}^{\text{global}}$, within a dynamic distance threshold $\delta_{i,j}^{\text{nnratio}} = \delta^{\text{sample}}(\|\text{Bbox}(p_{t,i}^{\text{local}})\|_2^{1/2} + \|\text{Bbox}(p_{t-1,j}^{\text{global}})\|_2^{1/2})/2$.

By calculating fusion similarity, each new local object is matched with a global object which has the highest similarity score. If no match is found with a similarity higher than $\delta^{\text{sim}}$, the local object will be treated directly as a new global object. For the two matching objects $o_{t,i}^{\text{local}}$ and $o_{t-1,j}^{\text{global}}$, the fused object $o_{t,j}^{\text{global}} = \langle f_{t,j}^{\text{global}}, p_{t,j}^{\text{global}} \rangle$. The fused semantic embedding $f_{t,j}^{\text{global}}$ is calculated as $f_{t,j}^{\text{global}} = (n f_{t-1,j}^{\text{global}} + f_{t,i}^{\text{local}})/(n+1)$, where $n$ represents the mapping times of $f_{t-1,j}^{\text{global}}$. The fused point cloud is directly taken as the union as $p_{t,j}^{\text{global}} = p_{t-1,j}^{\text{global}} \cup p_{t,i}^{\text{local}}$. After $t$ iterations, we construct global object list $O_t^{\text{global}} = \{o_{t,1}^{\text{global}}, \ldots, o_{t,N}^{\text{global}}\}$, where $o_{t,i}^{\text{global}} = \langle f_{t,i}^{\text{global}}, p_{t,i}^{\text{global}} \rangle$. The detailed algorithm is provided in Appendix A.3.

## 3.2 NODE CAPTION GENERATION

Given the global object list, our goal is to derive node captions from the information aggregated in Section 3.1. Object masks are first fed into the Vision-Language Model (VLM) to obtain initial captions (Section 3.2.1). We then render multi-view reconstructed point clouds to produce best-view re-shot images and perform re-shot guided uncertainty estimation by comparing captions from re-shot and original images (Section 3.2.2). Low-uncertainty objects directly form the object document, while high-uncertainty objects retrieve this document for caption refinement, yielding more accurate and robust 3D scene graphs (Section 3.2.3).

### 3.2.1 INITIAL CAPTION GENERATION

For each object in the global object list, we maintain the top-$k$ views with the highest segmentation confidence. Object-level crops from these top-$k$ views are fed into a VLM (Hurst et al., 2024) to obtain initial crop captions using the prompt "briefly describe the central object in the image in a few words." The initial crop captions for object $o_{t,i}^{\text{global}}$ are denoted as $c_{t,i} = \{c_1, \ldots, c_k\}$, which may be incorrect due to constrained viewpoints or occlusion as illustrated in the upper part of Figure 1.

### 3.2.2 RE-SHOT GUIDED UNCERTAINTY ESTIMATION

The initial crop captions for object $o_{t,i}^{\text{global}}$ may be unreliable due to constrained viewpoints and severe object occlusion. Relying solely on these captions could propagate noise into the subsequent aggregation of the object-level information. To mitigate this issue, we introduce a re-shot strategy that generates best-view re-shot images from object point clouds. Unlike multi-view crops from the original scene images, the reconstructed object-level point cloud $p$ contains only the object of interest, free from occlusion and background clutter. Since the point cloud can be observed from arbitrary viewpoints, we can render a 2D image from a perspective that maximally represents the object's geometry and appearance. This ensures that the resulting re-shot captions capture the most informative

object features. Conceptually, this approach is analogous to the viewpoint entropy (Vázquez et al., 2001) or best-view selection problem in computer vision, where the goal is to choose a view that maximizes information content.

Given an object point cloud $p$ with a maintained average camera position $v^{\text{avg}}$ from the images used to construct the point cloud, our goal is to render an optimal 2D view that best represents the object. To this end, we uniformly sample multiple candidate camera positions on a hemisphere centered at the object center $o$, and render the corresponding 2D re-shot images. To select the most informative view, we define a view quality score for each candidate position $c_i$ with three complementary terms:

$$S_{\text{vis}} = \frac{|p^{\text{visible}}|}{|p|}, \quad S_{\text{up}} = 1 - |v_i \cdot g|, \quad S_{\text{prior}} = \tfrac{1}{2}(1 + v_i \cdot f), \tag{5}$$

where $S_{\text{vis}}$ measures the visible ratio of points under hidden-point removal, $S_{\text{up}}$ evaluates the alignment of the view direction $v_i = o - c_i$ with the gravity vector $g$, and $S_{\text{prior}}$ enforces consistency with the prior direction $f = v^{\text{avg}} - o$. The overall score is then computed as $S_{\text{view}} = (1 - \alpha - \beta)S_{\text{vis}} + \alpha S_{\text{up}} + \beta S_{\text{prior}}$, with $\alpha$ and $\beta$ controlling the trade-off between uprightness and prior alignment. Based on the view quality score, we select the candidate view with the highest $S_{\text{view}}$ and render the corresponding 2D re-shot image $I^{\text{reshot}}$ (see Appendix A.4 for examples). The re-shot caption $c^{\text{reshot}}$ is then obtained from the VLM using the same prompt as in Section 3.2.1.

To quantify uncertainty, we compute the cosine similarity between the CLIP embeddings of the re-shot caption and the initial crop captions:

$$\{s_1, \ldots, s_k\} = \left\{ \cos\left(\text{CLIP}(c^{\text{reshot}}), \text{CLIP}(c_i)\right) \right\}_{i=1}^{k}. \tag{6}$$

We then perform clustering on the similarity scores and select the top-1 cluster $\{c_1, \ldots, c_l\}$, $\{s_1, \ldots, s_l\}$, $l \leq k$, which is considered the most reliable subset of initial captions for further refinement. The captions in this subset are aggregated into a single caption $\hat{c}$ using a Large Language Model (LLM) (Achiam et al., 2023) with a designed prompt. The corresponding similarity scores are averaged to obtain $\hat{s} = \frac{1}{l} \sum_{i=1}^{l} s_i$, where a higher $\hat{s}$ indicates stronger agreement among the captions, and thus lower uncertainty in $\hat{c}$.

### 3.2.3 OBJECT-LEVEL RAG

Based on the re-shot guided uncertainty estimation introduced in the previous section, we first rank all objects by their uncertainty scores $1 - \hat{s}$. We additionally apply a prompt to the VLM (Hurst et al., 2024) to filter out background objects via crops. The top-50% low-uncertainty objects are directly included in the object document for RAG, where each final caption $c$ is set to $\hat{c}$. For the remaining high-uncertainty objects, we perform refinement with the aid of contextual information. Specifically, a 3D position-based retriever retrieves the nearest object in the document, whose caption $c^{\text{env}}$ serves as augmented auxiliary context. In addition, we construct a composite image by concatenating the re-shot image (providing global context) with the crop image that yields the highest similarity score. This composite image, together with a text prompt containing $c^{\text{env}}$, is fed into a VLM (Hurst et al., 2024) to generate the refined caption $c$. The prompt is designed as: "The picture is stitched from the point cloud image and the RGB image of the same indoor object. There is a $c^{\text{env}}$ near the object. Briefly describe the object in the picture." Through the refinement process, we obtain a precise object list $O = \{o_1, \ldots, o_N\}$, where each object $o_i$ is represented as $o_i = \langle f_i, p_i, c_i \rangle$, consisting of its semantic embedding $f_i$, point cloud $p_i$, and precise node caption $c_i$.

### 3.3 EDGE CAPTION GENERATION

Following Gu et al. (2024), we estimate the spatial relationships among 3D objects to complete the scene graph. Given the set of 3D objects $O = \{o_1, \ldots, o_N\}$, we first compute their potential connectivity. Unlike Gu et al. (2024), which adopts a fixed NN-ratio threshold to build a dense graph and then prunes it with a minimum spanning tree (MST), we introduce a dynamic threshold similar to Equation 4, consistent with our dynamic downsample-mapping strategy, thus adapting the edge construction to varying point cloud densities.

For relationship captioning, Gu et al. (2024) employ a structured prompt that restricts the output to five predefined relation types and uses object captions and 3D locations as inputs. In contrast, we

extend this design by (i) expanding the relation space to eight categories (*"a on b," "b on a," "a in b," "b in a," "a part of b," "b part of a," "near," "none of these"*), and (ii) providing few-shot in-context examples to guide the model. This design ensures that the generated relationships are more expressive.

### 3.4 SCENE GRAPH REPRESENTATION

With the refined object list and edge captions, we formally define the final 3D scene graph as $\mathcal{G} = (O, E)$, where $O = \{o_1, \ldots, o_N\}$ denotes the set of objects and $E = \{e_{ij}\}$ the set of edges. Each object $o_i$ is represented as $o_i = \langle f_i, p_i, c_i \rangle$, consisting of its semantic embedding $f_i$, point cloud $p_i$, and precise caption $c_i$. Each edge $e_{ij}$ is represented as $e_{ij} = \langle o_i, o_j, r_{ij} \rangle$, where $r_{ij}$ is the discrete relation label selected from the predefined set of eight categories. This formulation yields a complete and interpretable 3D scene graph with both node- and edge-level semantic annotations (Figure 2).

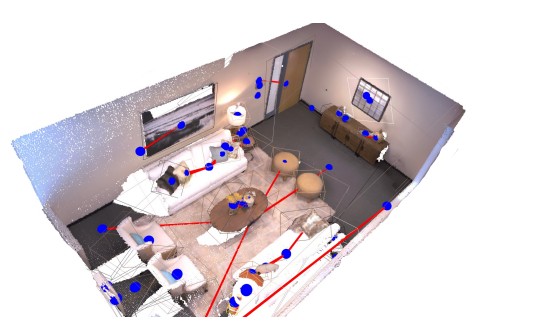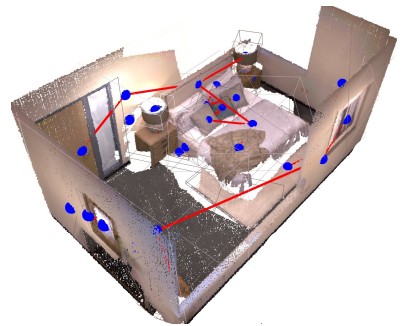

Figure 2: Visualization of our 3DSGs for Replica (Straub et al., 2019) Room 0 (left) and 1 (right). The blue points represent the objects and the red lines indicate the relationships between them.

## 4 EXPERIMENTS

### 4.1 IMPLEMENTATION DETAILS

For object segmentation, we use SAM($\cdot$) with the pretrained checkpoint `sam_vit_h_4b8939`. For encoding semantic embeddings, we use CLIP($\cdot$) with the `ViT-H-14` backbone pretrained on `laion2b_s32b_b79k`. For all purely text-based tasks, we employ `gpt-4-0613` as the LLM, while for vision-language tasks involving both images and text, we adopt `gpt-4o-mini` as the VLM. Regarding hyperparameters, we set the similarity threshold for object mapping to $\delta^{\text{sim}} = 0.45$, and the base voxel size for dynamic downsampling to $\delta^{\text{sample}} = 0.01$ meters. For the scoring function in re-shot guided uncertainty estimation, we set $\alpha = \beta = 0.2$. Other non-critical hyperparameters and prompts will be released with our code in the accompanying GitHub repository.

### 4.2 SCENE GRAPH CONSTRUCTION EVALUATION

We first evaluate the accuracy of the generated 3D scene graphs on Replica (Straub et al., 2019) dataset in Table 1. We compare our method against two baselines: ConceptGraphs (CG) and its variant ConceptGraphs-Detector (CG-D) (Gu et al., 2024). The open-vocabulary nature of our method makes automatic evaluation challenging. Therefore, following the protocol of ConceptGraphs (Gu et al., 2024), we resort to human evaluation. We recruited knowledgeable university students as annotators and randomly shuffled the basic evaluation units before distribution. For node evaluation, each unit consists of a point cloud, its mask images, and the predicted node caption, and annotators are asked to judge whether the caption is accurate. For edge evaluation, each unit includes two such node units along with the predicted edge caption and the whole scene point clouds, and annotators are similarly asked to assess the correctness of the relationship description.

Table 1: Performance comparison of 3D scene graph generation methods on Replica (Straub et al., 2019) dataset. Node precision, edge precision, duplicate objects and object/edge counts are evaluated through human annotation across multiple indoor scenes (room0-office4).

| scene | node prec. | | | valid objects | | | duplicates | | | edge prec. | | | valid edges | | |
|---|---|---|---|---|---|---|---|---|---|---|---|---|---|---|---|
| | Ours | CG | CG-D | Ours | CG | CG-D | Ours | CG | CG-D | Ours | CG | CG-D | Ours | CG | CG-D |
| room0 | **0.87** | 0.77 | 0.53 | 61 | 57 | 60 | 1 | 4 | 3 | **0.93** | 0.87 | 0.88 | 27 | 15 | 16 |
| room1 | **0.88** | 0.73 | 0.71 | 51 | 45 | 42 | 0 | 5 | 3 | **0.97** | 0.92 | 0.91 | 30 | 12 | 11 |
| room2 | **0.85** | 0.63 | 0.50 | 47 | 48 | 50 | 0 | 3 | 2 | **0.94** | 0.91 | 0.92 | 35 | 11 | 12 |
| office0 | **0.73** | 0.61 | 0.61 | 48 | 44 | 41 | 1 | 1 | 1 | **0.93** | 0.78 | 0.82 | 27 | 9 | 11 |
| office1 | **0.73** | 0.64 | 0.46 | 44 | 25 | 24 | 0 | 1 | 3 | **0.93** | 0.80 | 0.86 | 28 | 5 | 7 |
| office2 | **0.87** | 0.77 | 0.68 | 67 | 48 | 44 | 1 | 3 | 2 | **0.88** | 0.79 | 0.86 | 34 | 14 | 14 |
| office3 | **0.85** | 0.69 | 0.60 | 65 | 59 | 57 | 2 | 4 | 2 | **0.84** | 0.78 | 0.77 | 32 | 9 | 13 |
| office4 | **0.79** | 0.61 | 0.57 | 53 | 41 | 46 | 1 | 5 | 4 | **0.86** | 0.67 | 0.80 | 22 | 3 | 5 |
| Average | **0.82** | 0.68 | 0.58 | - | - | - | - | - | - | **0.91** | 0.82 | 0.85 | - | - | - |

From Table 1, our method consistently outperforms both ConceptGraphs (CG) and ConceptGraphs-Detector (CG-D) across most evaluation metrics. In terms of node precision, our method achieves an average score of 0.82, which is notably higher than CG (0.68) and CG-D (0.58), demonstrating the effectiveness of our re-shot guided uncertainty estimation in reducing noise during object caption aggregation. For edge precision, our method also attains the highest average score (0.91), surpassing CG (0.82) and CG-D (0.85), indicating that our structured prompt and refined relationship categories lead to more accurate and interpretable edge captions. In addition, our method substantially reduces duplicate predictions while maintaining a higher number of valid objects and edges, further confirming its robustness.

Overall, these results validate that our approach not only improves caption accuracy at both the node and edge levels but also enhances the reliability of the entire 3D scene graph construction pipeline.

### 4.3 SEMANTIC SEGMENTATION EVALUATION

We evaluate our dynamic downsample-mapping strategy on the closed-set Replica dataset (Straub et al., 2019) following the ground-truth construction and evaluation protocol of Gu et al. (2024); Jatavallabhula et al. (2023). Concretely, the ground-truth (GT) point clouds with per-point semantic labels are obtained as in ConceptGraph Gu et al. (2024): SemanticNeRF (Zhi et al., 2021) provides rendered RGB-D frames and 2D semantic masks, the masks are converted to one-hot per-pixel embeddings and fused into 3D via GradSLAM (Krishna Murthy et al., 2020), yielding the reference GT point cloud with per-point semantic annotations.

After performing the cross-image object mapping described in Section 3.1, our method produces object-level point clouds with fused semantic embeddings. To align predictions with GT categories, we map each GT object label to the predicted object whose fused semantic embedding has the highest cosine similarity with the CLIP text embedding of that GT label. Following the same protocol, for each point in the GT point cloud we compute its 1-NN in the predicted point cloud, compare the GT class with the predicted class of that 1-NN to build the confusion matrix, and compute the class-mean recall (mAcc) and the frequency-weighted mean intersection-overunion (f-mIOU).

We report our result combined with the results reported in Maggio et al. (2024); Gu et al. (2024) in Table 2. Our method shares the same overall matching pipeline with ConceptGraph, but significantly reduces the computational cost. Under the same base voxel size, our dynamic downsample-mapping strategy shortens the processing time by nearly two-thirds (e.g., from 6.65s/iter to 2.49s/iter when the voxel size is set to 0.01 in Replica (Straub et al., 2019) Room 0). Moreover, as shown in Table 2, our method achieves comparable or even superior accuracy, reaching the best mAcc score (40.67) while maintaining a competitive f-mIoU (35.65). This demonstrates that our approach not only accelerates the object mapping process but also preserves segmentation quality.

### 4.4 ABLATION STUDY

We conduct ablation studies to quantify the contribution of each component in our pipeline. Our full model (*Ours*) consists of dynamic downsample-mapping & fusion, re-shot guided uncertainty estimation, and node-level RAG with concatenated re-shot image prompts. We compare against

Table 2: Semantic segmentation experiments on Replica (Straub et al., 2019) dataset for object mapping and time evaluation. Baseline results are reported from Maggio et al. (2024); Gu et al. (2024). mAcc denotes class-mean recall and f-mIOU denotes frequency-weighted mean intersection-over-union reported from (Jatavallabhula et al., 2023).

| Method | mAcc | F-mIOU |
|---|---|---|
| MaskCLIP | 4.53 | 0.94 |
| Mask2former + Global CLIP feat | 10.42 | 13.11 |
| ConceptFusion | 24.16 | 31.31 |
| ConceptFusion + SAM | 31.53 | 38.70 |
| ConceptGraphs | 40.63 | 35.95 |
| ConceptGraphs-Detector | 38.72 | 35.82 |
| OpenMask3D | 39.54 | **49.26** |
| Clio-batch | 37.95 | 36.98 |
| Ours | **40.67** | 35.65 |



Figure 3: Ablation study quantifying the contribution of each pipeline component. Performance degradation is observed when removing reshot guided uncertainty estimation (w/o Reshot), RAG (w/o RAG), concatenated re-shot images (w/o Concat), or using random retrieval (Random RAG), confirming the complementary roles of all proposed components.

several variants: (i) removing re-shot guided uncertainty estimation (*w/o Reshot*), (ii) removing RAG (*w/o RAG*), (iii) applying random retrieval for RAG (*random-RAG*), and (iv) removing the concatenated re-shot image prompts (*w/o Concat*).

Experiments are conducted on Replica (Straub et al., 2019) dataset, following a protocol similar to Section 4.3. The ground-truth (GT) point clouds with per-point semantic labels are obtained as in ConceptGraphs (Gu et al., 2024). Different from Section 4.3, where fused semantic embeddings are directly compared with GT embeddings via cosine similarity, here we leverage the final node captions of predicted objects as semantic representations and employ GPT-4o as a semantic assigner. Concretely, for each GT object label, GPT-4o is prompted to determine the most likely corresponding predicted node based on the node captions, thereby enabling a more faithful evaluation under the open-vocabulary setting. After establishing this assignment, we compute 1-NN matching between GT and predicted point clouds, construct the confusion matrix, and report quantitative results in Figure 3. Detailed heatmaps for each individual scene are provided in Appendix A.5.

As shown in Figure 3, removing any component leads to a noticeable performance drop, confirming their complementary roles. In particular, removing re-shot guided uncertainty estimation (*w/o Reshot*) causes a drastic degradation in mF1 (14.66 vs. 30.78) and f-mIoU (35.56 vs. 54.26), underscoring its importance in filtering unreliable captions. Eliminating RAG (*w/o RAG*) reduces both precision and overall accuracy, while replacing it with random retrieval (*random-RAG*) further deteriorates performance, highlighting the necessity of semantic-aware retrieval. Removing the concatenated re-shot image (*w/o Concat*) also leads to lower recall and f-mIoU, suggesting that multi-view prompts alleviate viewpoint bias and enrich object descriptions. These results collectively demonstrate that all three proposed components contribute significantly to the robustness and accuracy of our framework.

# 5 CONCLUSION

In this work, we propose a 3DSG generation method named RAG-3DSG for more accurate and robust 3DSGs. We are the first to specifically address noise in cross-image information aggregation and incorporate an object-level RAG into 3DSGs for caption refinement. To evaluate our approach, we conduct experiments on Replica (Straub et al., 2019) dataset, which shows that RAG-3DSG significantly improves node captioning accuracy in 3DSG generation.

ETHICS STATEMENT

This work focuses on developing methods for open-vocabulary 3D scene graph generation using public dataset (Replica (Straub et al., 2019)). It does not involve human subjects, personal data, or sensitive information. Our work is intended solely for advancing robotics and embodied AI research in safe and beneficial contexts, such as robot navigation and manipulation. We encourage responsible use aligned with ethical research practices.

REPRODUCIBILITY STATEMENT

We have made extensive efforts to ensure reproducibility. All implementation details, including pipline architectures, foundation models, and hyperparameters, are provided in the main text and appendix. The datasets we used (Replica (Straub et al., 2019)) are publicly available. Our code, along with evaluation scripts, will be released on GitHub to facilitate replication. We also describe the evaluation procedure in detail, so that results can be reproduced independently.

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

## A APPENDIX

### A.1 LARGE LANGUAGE MODEL USAGE

Large Language Models are utilized in this work to aid and polish writing, specifically to improve clarity, grammatical accuracy, and overall readability of the manuscript. All technical content, scientific contributions, and research findings remain entirely the work of the authors, who take full responsibility for the entire content of this paper.

### A.2 DYNAMIC DOWNSAMPLE EXAMPLES

We provide detailed examples of our dynamic downsampling strategy for indoor 3D pointclouds in Table 3. The voxel size for each object is computed based on its spatial extent using Equation 1 in Section 3.1.2.

### A.3 PSEUDO CODE FOR OBJECT MAPPING

For clarity and reproducibility, we provide the pseudo code of our incremental 3D object mapping algorithm in Algorithm 1. This algorithm describes the step-by-step procedure of updating global object list from local object lists of incoming frames, including point cloud integration and semantic refinement. It serves as a concise summary of the mapping pipeline presented in the main paper.

### A.4 RE-SHOT IMAGES

Figure 4 shows some examples of re-shot images automatically selected by our method. As discussed in Section 3.2.2, the initial image crops of an object may suffer from occlusions or constrained viewpoints, leading to unreliable captions and noisy semantics. To address this issue, we propose a re-shot strategy that leverages object-level point clouds to render new views from arbitrary perspectives, ensuring that the essential geometry and appearance of the object are faithfully captured. This approach effectively eliminates occlusion and viewpoint constraints inherent in the original images.

### A.5 ALL HEATMAPS OF ABLATION STUDY ON REPLICA

As shown in Figure 5, we present the heatmaps of semantic segmentation metrics across different scenes and ablation settings.

### A.6 CASE STUDY

Figure 6 demonstrates a concrete example of our re-shot guided uncertainty estimation when LLaVA (Liu et al., 2023) is the VLM. Multiple crop images incorrectly identify the object as"vase" due to occlusions and constrained viewing angles. In contrast, the re-shot image (top-left) provides a correct caption from an optimal viewpoint selected from our algorithm, avoiding the viewpoint limitations and occlusions that plague the original crop images. The low similarity scores flag these crop captions as unreliable, triggering our object-level RAG caption refinement process.

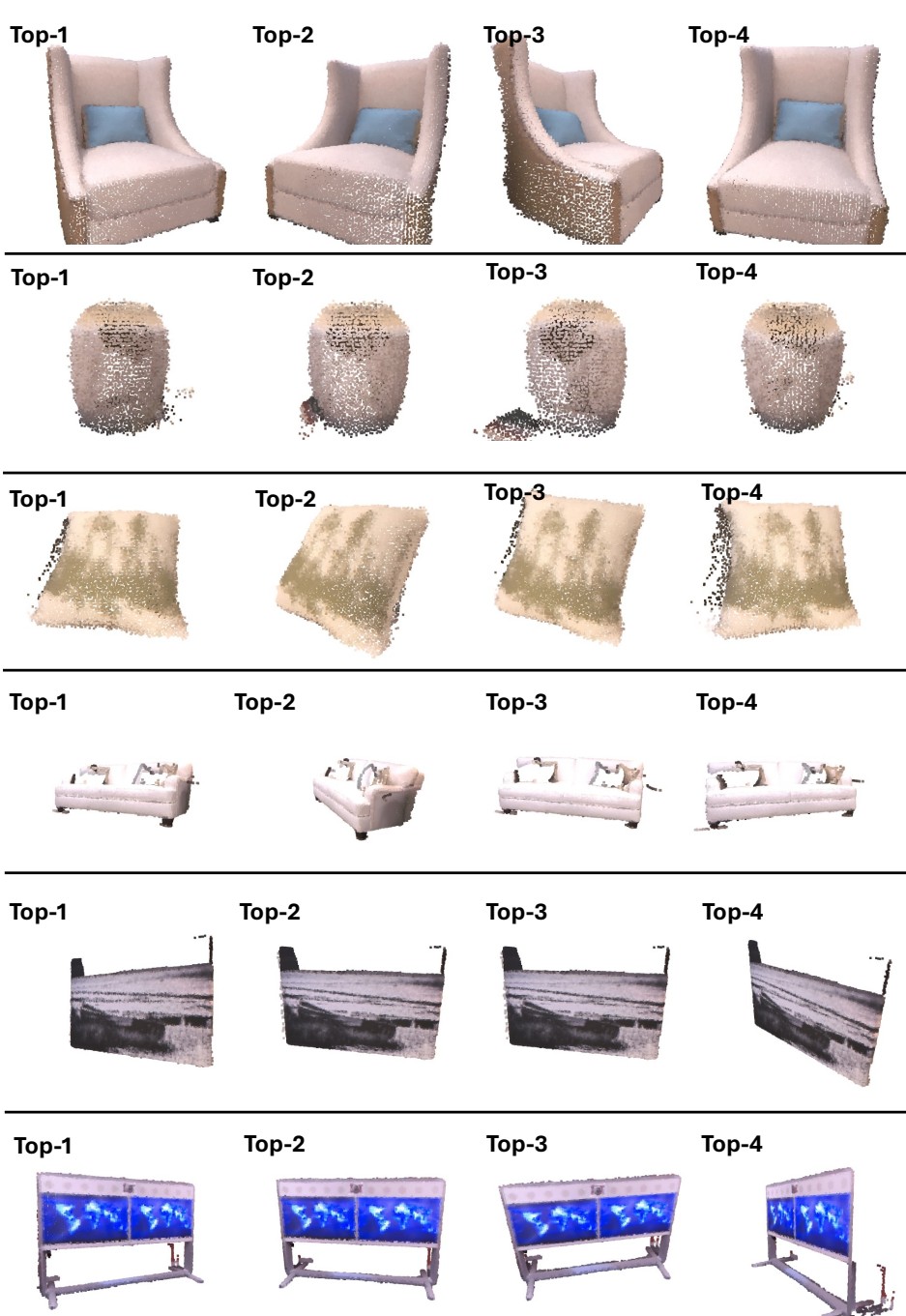

Figure 4: Examples of re-shot images automatically selected by our method. For each object category, we present the top-4 rendered views ranked by our re-shot scoring strategy. From top to bottom: armchair, vase, cushion, sofa, artwork, and TV stand.

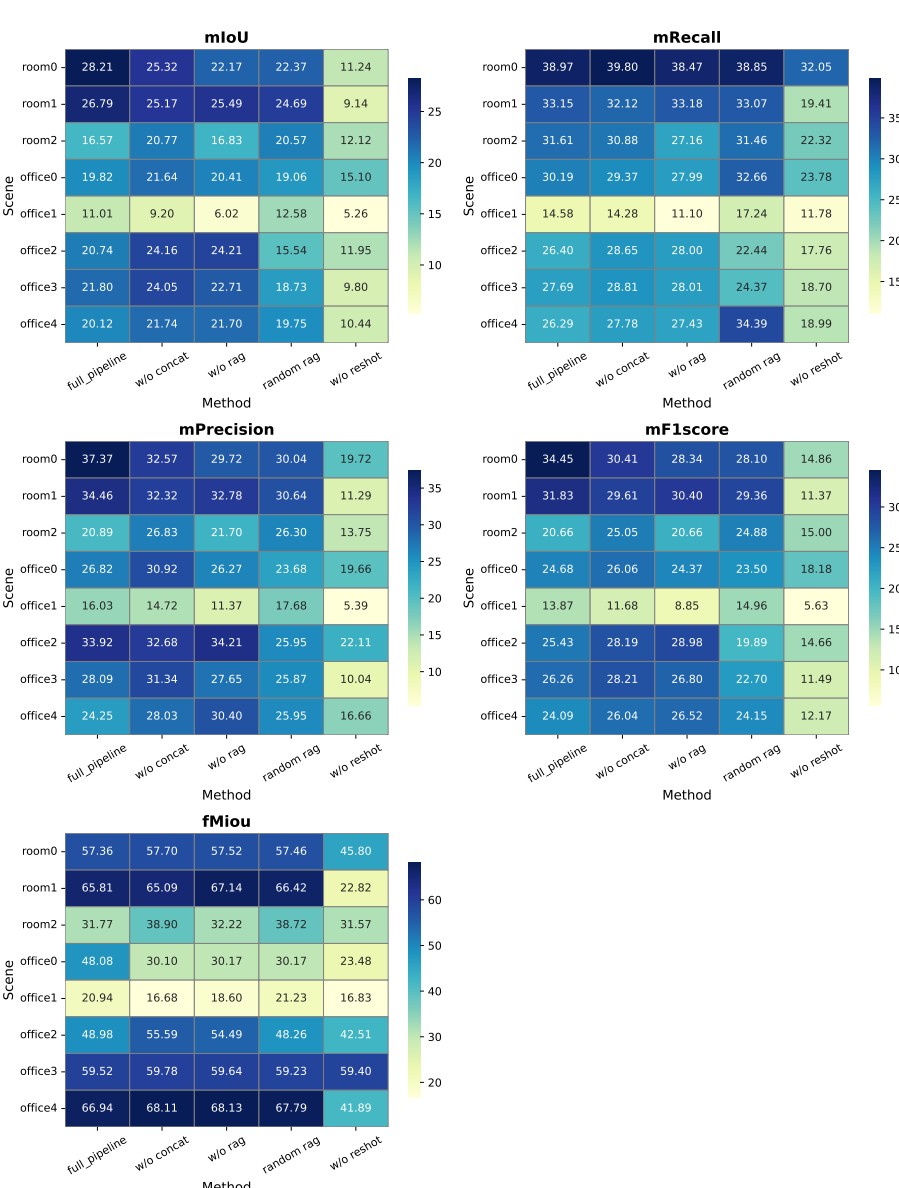

Figure 5: Heatmaps of semantic segmentation performance across scenes for different ablation settings. Metrics include mIoU, mRecall, mPrecision, mF1score, and frequency-weighted mIoU (fMiou). The full pipeline serves as the baseline, while "w/o concat", "w/o rag", "random rag", and "w/o reshot" show the impact of removing or modifying individual components.

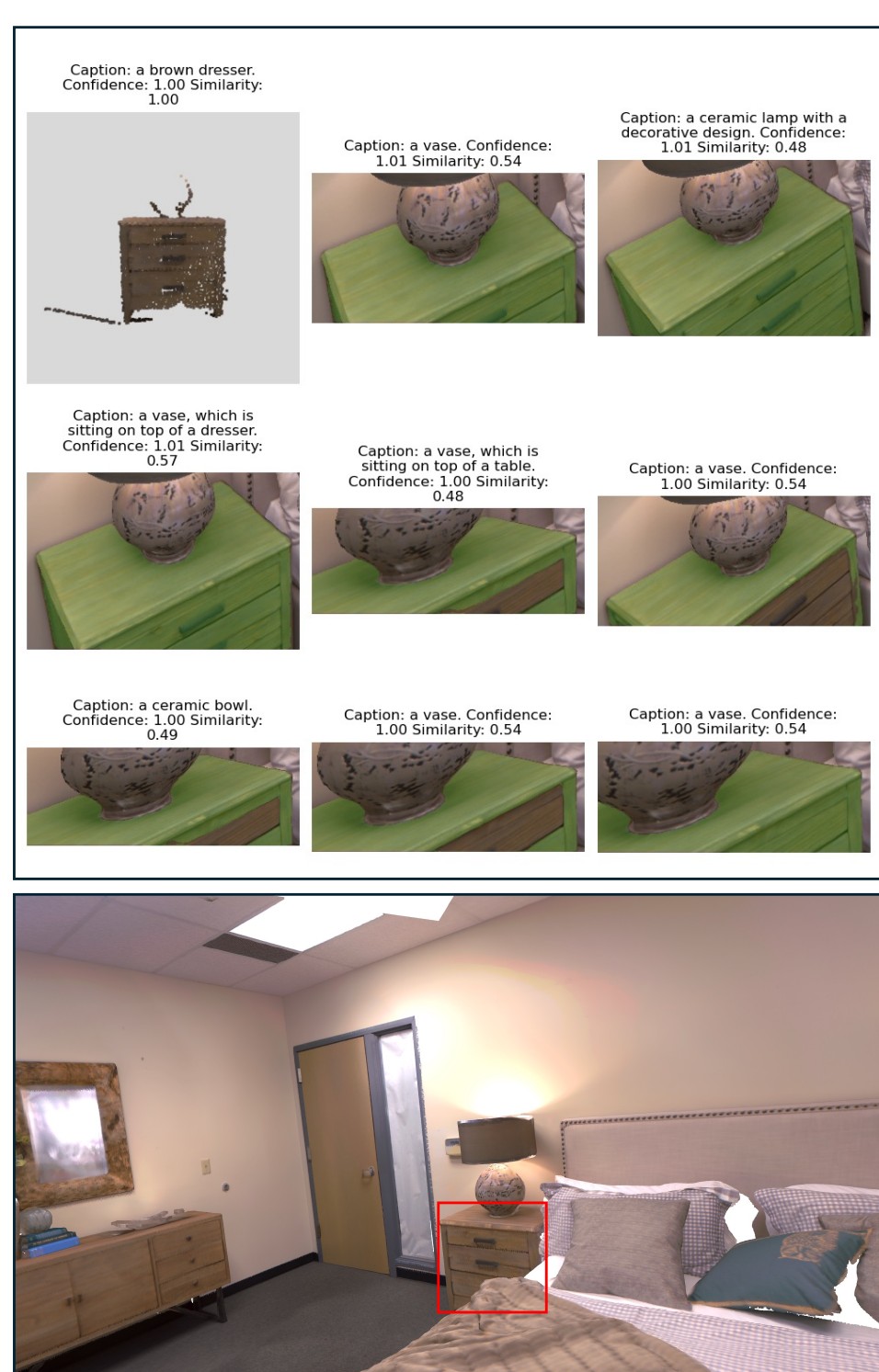

Figure 6: Case study of re-shot guided uncertainty estimation using LLaVA (Liu et al., 2023) as VLM. Top panel: Re-shot image with caption (top-left) and crop images from original viewpoints with their captions and similarity scores to the re-shot image. Bottom panel: Complete scene reconstruction via GradSLAM (Krishna Murthy et al., 2020) with the target object highlighted in red bounding box. The crop images consistently misidentify the object as "vase" due to occlusions and constrained viewpoints, resulting in low similarity scores that indicate high uncertainty.

Table 3: Examples of dynamic voxel sizes for different objects with base voxel size $\delta^{\text{sample}} = 0.01$m.

| Object Category | Typical Size (L×W×H) | Diagonal Length | Voxel Size | Reduction Factor |
|---|---|---|---|---|
| Small Objects | | | | |
| Coffee cup | $0.08 \times 0.08 \times 0.12$ | 0.165 m | 0.004 m | 2.5× |
| Smartphone | $0.15 \times 0.07 \times 0.008$ | 0.166 m | 0.004 m | 2.5× |
| Computer mouse | $0.12 \times 0.06 \times 0.04$ | 0.14 m | 0.0037 m | 2.7× |
| Medium Objects | | | | |
| Monitor | $0.55 \times 0.32 \times 0.05$ | 0.638 m | 0.008 m | 1.25× |
| Chair | $0.60 \times 0.55 \times 0.85$ | 1.177 m | 0.011 m | 0.9× |
| Desk | $1.20 \times 0.60 \times 0.75$ | 1.537 m | 0.012 m | 0.8× |
| Large Objects | | | | |
| Sofa | $2.00 \times 0.90 \times 0.85$ | 2.352 m | 0.015 m | 0.67× |
| Dining table | $2.50 \times 1.20 \times 0.75$ | 2.873 m | 0.017 m | 0.59× |
| Bookshelf | $0.80 \times 0.30 \times 2.20$ | 2.36 m | 0.015 m | 0.67× |
| Extra Large Objects | | | | |
| Wall | $10.0 \times 0.20 \times 3.0$ | 10.44 m | 0.032 m | 0.03× |
| Floor | $10.0 \times 10.0 \times 0.05$ | 10.00 m | 0.0316 m | 0.03× |

---

**Algorithm 1** 3D Object Fusion with dynamic threshold

---

**Require:** Local object list $O_t^{\text{local}}$ for frame $t$, similarity threshold $\delta^{\text{sim}}$, sample threshold $\delta^{\text{sample}}$
**Ensure:** Global object list $O_t^{\text{global}}$

1: Initialize global object list: $O_1^{\text{global}} = O_1^{\text{local}}$
2: **for** $t = 2$ to $T$ **do**
3:    **for** each local object $o_{t,i}^{\text{local}} = (f_{t,i}^{\text{local}}, p_{t,i}^{\text{local}}) \in O_t^{\text{local}}$ **do**
4:       best_match = None
5:       max_similarity = 0
6:       **for** each global object $o_{t-1,j}^{\text{global}} = (f_{t-1,j}^{\text{global}}, p_{t-1,j}^{\text{global}}) \in O_{t-1}^{\text{global}}$ **do**
7:          // Calculate semantic similarity
8:          $\theta_{\text{semantic}}(i,j) = (f_{t,i}^{\text{local}})^T f_{t-1,j}^{\text{global}} / 2 + 1/2$
9:          // Calculate spatial similarity
10:         $\delta_{i,j}^{\text{nnratio}} = \delta^{\text{sample}} \cdot (\|\text{Bbox}(p_{t,i}^{\text{local}})\|_2^{1/2} + \|\text{Bbox}(p_{t-1,j}^{\text{global}})\|_2^{1/2})/2$
11:         $\theta_{\text{spatial}}(i,j) = \text{dnnratio}(p_{t,i}^{\text{local}}, p_{t-1,j}^{\text{global}})$ {Using threshold $\delta_{i,j}^{\text{nnratio}}$}
12:         // Calculate fusion similarity
13:         $\theta(i,j) = \theta_{\text{semantic}}(i,j) + \theta_{\text{spatial}}(i,j)$
14:         **if** $\theta(i,j) >$ max_similarity **then**
15:            max_similarity $= \theta(i,j)$
16:            best_match $= j$
17:         **end if**
18:       **end for**
19:       **if** max_similarity $> \delta^{\text{sim}}$ and best_match $\neq$ None **then**
20:          // Fuse with matched global object
21:          $j = $ best_match
22:          $n = \text{mapping\_times}(f_{t-1,j}^{\text{global}})$
23:          $f_{t,j}^{\text{global}} = (n \cdot f_{t-1,j}^{\text{global}} + f_{t,i}^{\text{local}})/(n+1)$
24:          $p_{t,j}^{\text{global}} = p_{t-1,j}^{\text{global}} \cup p_{t,i}^{\text{local}}$
25:          $o_{t,j}^{\text{global}} = (f_{t,j}^{\text{global}}, p_{t,j}^{\text{global}})$
26:       **else**
27:          // Create new global object
28:          Add $o_{t,i}^{\text{local}}$ to $O_t^{\text{global}}$ as a new global object
29:       **end if**
30:    **end for**
31: **end for**
32: **return** $O_t^{\text{global}} = \{o_{t,1}^{\text{global}}, \ldots, o_{t,N}^{\text{global}}\}$ where $o_{t,i}^{\text{global}} = (f_{t,i}^{\text{global}}, p_{t,i}^{\text{global}})$

---

