# OpenReview forum: "RAG-3DSG: Enhancing 3D Scene Graphs with Re-Shot Guided Retrieval-Augmented Generation"
_ICLR.cc/2026/Conference — ICLR 2026 Conference Withdrawn Submission_

### Official Review · Reviewer_Ym3X · 2025-10-21

**Soundness:** 3
**Presentation:** 2
**Contribution:** 3
**Rating:** 2
**Confidence:** 3

**Summary:**

This paper introduces RAG-3DSG, a framework for generating open-vocabulary 3D Scene Graphs. The authors identify that existing methods suffer from noise and inaccuracies introduced during cross-image aggregation, often caused by poor viewpoints and occlusions. To address this, they propose a multi-stage pipeline. Key contributions include: (1) a "re-shot" strategy that renders reconstructed object point clouds to obtain a best-view image, which is used to estimate the uncertainty of initial captions generated from cropped images; (2) an object-level Retrieval-Augmented Generation (RAG) mechanism where captions for high-uncertainty objects are refined using contextual information from nearby low-uncertainty objects ; and (3) a dynamic downsampling strategy for object point clouds to accelerate the cross-image mapping process. Experiments conducted on the Replica dataset show that RAG-3DSG outperforms existing methods in node and edge captioning accuracy.

**Strengths:**

- The proposed object-level RAG framework is a clever way to leverage the scene's internal structure for self-correction. Treating confident, low-uncertainty objects as a knowledge base and using them to refine the descriptions of ambiguous, high-uncertainty objects is a effective strategy.
- The core idea of using a re-shot image, rendered from an aggregated object point cloud, is a novel and intuitive approach to mitigate issues of occlusion and constrained viewpoints. By generating an image from an optimal, unobstructed perspective, the method creates a reliable reference caption to gauge the quality of captions from original, potentially noisy views.

**Weaknesses:**

- The experiments are performed exclusively on the synthetic Replica dataset. While this is a standard benchmark, it consists of few clean, high-quality indoor environments. The method's effectiveness and robustness remain unproven on real-world datasets like ScanNet or Matterport3D. The re-shot performance on sparse or noisy data from real-world scans is a major open question.
- The reported performance on Table 2 are very marginal, if the computational cost is the main goal, it has to be reported along with performance metrics.
- The proposed pipeline seems computationally intensive and operates purely offline.

**Questions:**

- The evaluation has to be more comprehensive; can you also perform the evaluation on ScanNet dataset at least?
- Can you explicitly report the computational cost compared to the previous methods?
- There are several works that also focuses on computational cost for 3D semantic mapping (e.g., OpenFusion, etc.). It would be better to include those in the related work.

---

### Official Review · Reviewer_qFWs · 2025-10-24

**Soundness:** 2
**Presentation:** 3
**Contribution:** 2
**Rating:** 4
**Confidence:** 4

**Summary:**

This paper presents RAG-3DSG, a novel framework for open-vocabulary 3D scene graph (3DSG) generation. The authors argue that existing approaches suffer from noise during cross-view information aggregation, caused by limited viewpoints and occlusions, which degrades node (object) recognition accuracy. To address this limitation, the core contribution is a “re-shot guided uncertainty estimation” mechanism. It renders a re-shot image from the optimal viewpoint of each reconstructed object point cloud and compares this rendering with the original image crop to quantify the reliability of the object’s description. For objects with high uncertainty, the framework further employs retrieval-augmented generation (RAG) to retrieve contextual information from already identified, high-confidence objects within the scene, thereby refining the description of the uncertain object. Additionally, a dynamic downsampling strategy is proposed to improve computational efficiency. Experiments on the Replica dataset demonstrate that the proposed method achieves higher precision for both nodes and edges compared with the baseline, ConceptGraphs.

**Strengths:**

The paper accurately identifies a critical challenge in existing open-vocabulary 3DSG methods—namely, that semantic noise introduced during multi-view information aggregation, due to suboptimal viewpoints and occlusions, significantly degrades the quality of the resulting scene graphs. This is a widespread and pressing issue in real-world robotic applications.

The proposed “re-shot guided uncertainty estimation” is a highly novel and elegant idea. It creatively leverages the advantages of 3D reconstruction to provide an internal and self-consistent reference for assessing the quality of 2D visual observations, without requiring any external annotations. The integration of this uncertainty estimate with a retrieval-augmented generation (RAG) framework offers a logically coherent solution for handling difficult samples (i.e., objects with high uncertainty), making the overall approach both innovative and appealing.

The paper is well-structured and presents a clear, logical flow. In particular, Figure 1 effectively illustrates the limitations of existing methods and the full pipeline of RAG-3DSG, enabling readers to quickly grasp the core mechanism. The individual components of the method are also described in sufficient detail.

**Weaknesses:**

Limited Experimental Validation
· Single dataset: All experiments are conducted exclusively on Replica—a near-noise-free, synthetic benchmark. The paper’s central claim, however, is robustness to “limited viewpoints, occlusions, and adverse imaging conditions”, which are far more severe in real-world RGB-D streams captured by commodity sensors (e.g., Kinect, RealSense). No evidence is provided on more challenging, real-world datasets such as ScanNet, Matterport3D, or data collected by the authors themselves, substantially weakening the generality and practical relevance of the conclusions.
· Reliance on high-quality reconstruction: The re-shot mechanism presupposes accurate multi-view point-cloud fusion. In Replica’s idealized setting, reconstructions are almost perfect; in real scenes containing sensor noise, dynamic objects, and illumination drift, severe occlusions yield sparse or incomplete object point clouds. The resulting re-shot images will themselves be degraded, rendering the uncertainty estimate unreliable and the subsequent RAG step ineffective. The paper neither analyzes nor mitigates this potential failure mode.

Insufficient Details on Human Evaluation
Many key metrics (e.g., node/edge precision) rely on human judgments, yet the paper only states that “knowledgeable undergraduate students were recruited as annotators.” Critical experimental details are missing:

- Inter-annotator agreement: No κ or α scores are reported. Without a consistency check, readers cannot assess the reliability of the human labels—the de-facto ground truth for this study.
- Annotation protocol: The precise definition of “accurate” is not provided. Are “a wooden table” and “a brown desk” both considered correct for the same object? Where is the boundary between acceptable paraphrase and semantic error?

Omitting these standard reporting elements undermines the objectivity and credibility of the evaluation, a major weakness for any work whose conclusions hinge on human-rated data.

**Questions:**

1: Regarding human evaluation, could you provide more details, especially the number of annotators, the inter-annotator agreement score, and the specific guidelines for determining the accuracy of node/edge descriptions?
2:The Ablation Study (Figure 3) shows the largest performance drop after removal of the "w/o Reshot" module, which strongly demonstrates its importance. But the performance drop after removal of RAG (w/o RAG) appears to be less significant than expected. Does this mean that the proportion of high-uncertainty objects in the Replica dataset that actually benefit from RAG is not high?

---

### Official Review · Reviewer_R5A5 · 2025-10-29

**Soundness:** 2
**Presentation:** 2
**Contribution:** 2
**Rating:** 2
**Confidence:** 5

**Summary:**

The paper introduces RAG-3DGS, a RAG-enhanced open-vocabulary 3d scene graph generation framework that improve object caption and aggregation via object-level online RAG. The methods combine pretrained open-vocabulary vision foundation models with a new cross-image object association mechanism and propose a novel object-level RAG process to redine the node and edge caption results. Extensive experimental results demonstrate the superiority of the proposed RAG-3DGS.

**Strengths:**

The authors introduce a dedicated object-level RAG construction to improve node and edge caption in 3D scene graph. The designs are reasonable and well-supported by and experimental results.

**Weaknesses:**

1.The writing and organization are unclear. There are many components and steps in the proposed method, but the figure.1 and the text are not enough to explain the technique details, e.g., best render view in sec.3.2.2, it is hard to understand from text.

2.The authors do not provide a thorough evaluation of the effectiveness of the proposed method. The core of the proposed method is the caption refinement based on object-level RAG, the authors should provide a baseline method with designed prompt that take the surrounding object and background into consideration to output object and relationship caption in once, but without RAG.

3.The novelty of cross-image association mechanism is limited, and the superioritis of running time/memory is not analyzed in experiments.

4.The explanation of used prompts is unclear. The authors should show the prompts used in appendix?

**Questions:**

1.What is the meaning of “redundant surface density” in the abstract, which seems not mentioned in main text?

2.What is the meaning of dynamic distance threshold to compute spatial similarity in Eq.4, why just set it with a constant?

3.In Sec. 3.2.1, the prompt used to generate initial caption are “briefly describe the central object in the image in a few words”, what if the object is not at the central of image? What if we annotate the object with bbox on image, and prompt VLM with ““briefly describe the object annotateed with bbox in the image in a few words”.

4.How exactly the edge is constructed, which is not clear in Sec. 3.3.

---

### Official Review · Reviewer_mkP2 · 2025-11-01

**Soundness:** 2
**Presentation:** 2
**Contribution:** 1
**Rating:** 2
**Confidence:** 5

**Summary:**

Authors propose RAG-3DSG to mitigate aggregation noise through reshot guided uncertainty estimation and support object-level Retrieval-Augmented Generation (RAG) via reliable low-uncertainty objects. Authors propose a dynamic downsample-mapping strategy to accelerate cross-image object aggregation with adaptive granularity. Experiments on Replica dataset demonstrate that RAG-3DSG significantly improves node captioning accuracy in 3DSG generation while reducing the mapping time by two-thirds compared to the vanilla version.

**Strengths:**

- Under the same base voxel size, our dynamic downsample-mapping strategy shortens the processing time by nearly two-thirds (e.g., from 6.65 s/iter to 2.49 s/iter when the voxel size is set to 0.01 in Replica).

- The proposed method achieves comparable or even superior accuracy, reaching the best mAcc score (40.67) while maintaining a competitive f-mIoU (35.65). This demonstrates that our approach not only accelerates the object mapping process but also preserves segmentation quality.

**Weaknesses:**

1) The superiority of the developed approach over its SOTA counterparts is unconvincing. Judging by Table 2, well-known methods from ConceptGraphs and OpenMask3D outperform the presented method on a number of metrics.

2) Furthermore, the comparison with existing methods is incomplete. For example, there are other effective methods for solving the problem of open-vocabulary scene graph generation, such as Beyond Bare Queries [1].
[1] Linok, S., Zemskova, T., Ladanova, S., Titkov, R., Yudin, D., Monastyrny, M., & Valenkov, A. Beyond Bare Queries: Open-vocabulary Object Grounding with 3D Scene Graph. In 2025 IEEE International Conference on Robotics and Automation (ICRA) (pp. 13582–13589). IEEE.

3) Testing only on the small and high-quality Replica dataset is insufficient; it is also necessary to demonstrate the results of the approach on more photorealistic datasets such as ScanNet (Sr3D+/Nr3D) [1] and 3DSSG [2].

[1] P. Achlioptas, A. Abdelreheem, F. Xia, M. Elhoseiny, and L.J. Guibas, “ReferIt3D: Neural listeners for fine-grained 3D object identification in real-world scenes,” in 16th European Conference on Computer Vision (ECCV), 2020

[2] Wald, J., Dhamo, H., Navab, N., & Tombari, F. (2020). Learning 3D semantic scene graphs from 3D indoor reconstructions. In Proceedings of the IEEE/CVF Conference on Computer Vision and Pattern Recognition (pp. 3961–3970).

4) Judging by Figure 1, the proposed RAG-3DSG framework is a modular approach; however, the rationale for the choice and evaluation of the impact of its individual hyperparameters on the final result is not provided. The ablation study is incomplete. For example, using LLM to generate graph edges seems to be a very resource- and time-intensive procedure.

**Questions:**

1) How does the inference performance (inference time) of the developed approach compare with the existing methods considered (not only ConceptGraphs)?

---

### Note · Authors · 2025-11-27

I have read and agree with the venue's withdrawal policy on behalf of myself and my co-authors.